# The Long Non-Coding RNA H19 Drives the Proliferation of Diffuse Intrinsic Pontine Glioma with H3K27 Mutation

**DOI:** 10.3390/ijms22179165

**Published:** 2021-08-25

**Authors:** David Roig-Carles, Holly Jackson, Katie F. Loveson, Alan Mackay, Rebecca L. Mather, Ella Waters, Massimiliano Manzo, Ilaria Alborelli, Jon Golding, Chris Jones, Helen L. Fillmore, Francesco Crea

**Affiliations:** 1Cancer Research Group, School of Life, Health and Chemical Sciences, The Open University, Milton Keynes MK7 6AA, UK; david.roig-carles1@open.ac.uk (D.R.-C.); holly.jackson@open.ac.uk (H.J.); rebecca.mather90@gmail.com (R.L.M.); ella.waters@open.ac.uk (E.W.); jon.golding@open.ac.uk (J.G.); 2School of Pharmacy and Biomedical Sciences, University of Portsmouth, Portsmouth PO1 2UP, UK; katie.loveson@port.ac.uk (K.F.L.); helen.fillmore@port.ac.uk (H.L.F.); 3Division of Molecular Pathology, The Institute of Cancer Research, London SW7 3RP, UK; Alan.MacKay@icr.ac.uk (A.M.); Chris.Jones@icr.ac.uk (C.J.); 4Institute of Pathology, University Hospital Basel, 4031 Basel, Switzerland; massimiliano.manzo@usb.ch (M.M.); ilaria.alborelli@usb.ch (I.A.)

**Keywords:** paediatric glioma, DIPG, diffuse midline glioma, lncRNA, epigenetics, brain cancer

## Abstract

Diffuse intrinsic pontine glioma (DIPG) is an incurable paediatric malignancy. Identifying the molecular drivers of DIPG progression is of the utmost importance. Long non-coding RNAs (lncRNAs) represent a large family of disease- and tissue-specific transcripts, whose functions have not yet been elucidated in DIPG. Herein, we studied the oncogenic role of the development-associated *H19* lncRNA in DIPG. Bioinformatic analyses of clinical datasets were used to measure the expression of *H19* lncRNA in paediatric high-grade gliomas (pedHGGs). The expression and sub-cellular location of *H19* lncRNA were validated in DIPG cell lines. Locked nucleic acid antisense oligonucleotides were designed to test the function of *H19* in DIPG cells. We found that *H19* expression was higher in DIPG vs. normal brain tissue and other pedHGGs. *H19* knockdown resulted in decreased cell proliferation and survival in DIPG cells. Mechanistically, *H19* buffers *let-7* microRNAs, resulting in the up-regulation of oncogenic *let-7* target (e.g., *SULF2* and *OSMR*). *H19* is the first functionally characterized lncRNA in DIPG and a promising therapeutic candidate for treating this incurable cancer.

## 1. Introduction

Malignant brain tumours are a leading cause of death in paediatric patients. Diffuse intrinsic pontine glioma (DIPG) is a type of paediatric high-grade glioma (pedHGG) originating in the brainstem and affecting children with a median age of six to seven years [1,2]. DIPGs are highly infiltrative and belong to the fibrillary astrocytoma family, where lesions are classified as either WHO grade III or IV [3]. At the molecular level, substitution of lysine at the 27 position of the histone 3 locus with methionine (*H3K27M* at either *H3.1* or *H3.3*) has been suggested to drive the oncogenesis of DIPGs [4]. *H3K27M* substitution is identified in approximately 80% of histologically confirmed DIPGs. The World Health Organization has recently classified DIPGs in the broader category of diffuse midline gliomas (DMGs) with a *H3K27M* mutation [2,5,6,7]. Another frequent mutation of the H3 locus, GR4R, is more prevalent in pedHGGs located in the cerebral hemispheres [8]. The surgical removal of DIPGs is almost impossible due to their infiltrative nature and their location. Currently, radiotherapy is the standard treatment for this malignancy; despite this treatment, the median survival for DIPG patients is 9–11 months [9,10]. Therefore, novel therapeutic targets are necessary to improve the prognosis of these paediatric patients.

There are approximately 60,000 long non-coding RNAs (lncRNAs) in the human genome; these transcripts are defined as non-coding transcripts longer than 200 nucleotides. Some lncRNAs are highly expressed under pathological conditions and regulate key aspects of tumour progression, invasion, and metastasis [11,12]. Previous studies have identified several lncRNAs associated with DIPG progression [13,14]. However, no lncRNAs have been functionally characterised in this malignancy.

*H19* (gene ID: 283120) is an extensively studied lncRNA, whose aberrant expression has been linked to alterations during foetal development [15]. The oncogenic role of *H19* has been described in several malignancies, including adult gliomas [16,17,18]. Mechanistically, *H19* expression is triggered by hypoxia; this lncRNA can affect a wide range of hub genes, including microRNAs (miRNAs) and mRNAs [19,20]. However, the function of H19 in pedHGG still remains largely unknown.

Herein, we investigated the clinical relevance of *H19* in DIPG cells and tested H19 targeting for halting DIPG proliferation.

## 2. Results

### 2.1. H19 Is Up-Regulated in DIPG Tissue

To confirm the expression of *H19* lncRNA in DIPG tissue, we analysed open-access clinical datasets of pedHGGs and normal brain tissue. Bioinformatic analyses confirmed that *H19* levels are significantly increased in the DIPG tissue of the “Allis-45-custom-ilmnht12v4” dataset (*p* < 0.01, Figure 1a) whereas an increasing trend was observed in “Paugh-37-MAS5.0-u133p2” (Figure 1b). Microarray data from Paediatric Cbioportal (https://pedcbioportal.kidsfirstdrc.org/, accessed 30 June 2020) demonstrated significantly higher *H19* expression in the brainstem than in hemispheric pedHGGs (*p* < 0.01). (Figure 1c). We then explored the influence of mutations in the *H3* genes (*H3K27M* and WT) on *H19* expression; this analysis revealed that H3K27M-bearing pedHGGs express higher levels of *H19* compared to pedHGGs bearing the wild-type *H3* gene (*p* < 0.01) (Figure 1d). Histological classification of pedHGG samples showed that DIPG tissues express higher *H19* levels than other pedHGGs, including anaplastic astrocytoma and glioblastoma (*p* < 0.01) (Figure 1e). Overall survival analysis did not identify significant differences between high- and low-*H19*-expressing DIPG groups in the CBioportal dataset (median survival was 9.8 and 11.1 months, respectively) (Figure 1f). Analysis of a single-cell RNA sequencing dataset from the developing human midbrain showed that *H19* is most strongly expressed by oligodendrocyte precursor cells (OPCs), which are thought to be the cells of origin of DIPGs [21,22] (Figure 1g). All of these results indicate that *H19* is highly and selectively up-regulated in DIPGs with *H3K27M* mutation. Hence, we decided to study the cellular function and mechanism of action of this lncRNA in DIPG-H3K27M.

### 2.2. H19 Is Required for DIPG Cell Viability but Not Migration

To investigate the functional role of *H19*, expression of this transcript was analysed in a panel of human DIPG cell lines and normal astrocytes. The malignant cell lines showed higher levels of *H19* lncRNA expression compared to normal astrocytes. VUMC-DIPG-A cells expressed the highest levels of *H19* (Figure 1a). Biologically, the VUMC-DIPG-A cell line bears the H3.3K27M mutation, whereas SU-DIPG-IV cells carry the H3.1K27M mutation (Appendix A). Cell fractionation assays demonstrated that *H19* lncRNA is ~3-fold enriched in the cytoplasm of both VUMC-DIPG-A and SU-DIPG-IV cells compared to the nucleus (Figure 2b).

To understand the cellular function of *H19*, we performed gene silencing studies on VUMC-DIPG-A cells using locked-nucleic acid (LNA) antisense oligonucleotides. Three LNAs were designed and tested in this cell line. Two days post-transfection, *H19* expression vs. the control was 0.47 ± 0.37 with LNA1, 0.16 ± 0.09 with LNA2 (*p* < 0.05), and 2.04 ± 2.59 with LNA3 (Figure 2c). LNA2 was therefore selected for cell proliferation assays. We found that cell numbers failed to increase in the presence of LNA2, which instead induced a significant and durable reduction in the number of cells on days 4 and 7 (*p* < 0.01 and *p* < 0.05, respectively) (Figure 2d). VUMC-DIPG-A cells were also imaged using bright-field microscopy, which showed morphological changes such as detached cells indicative of cell death upon exposure to LNA2 (Figure 2e). We also observed that LNA2 induced a dose-dependent inhibition of DIPG cell numbers, with an estimated IC_50_ of 52.4 ± 1.4 nM on day 4, measured by trypan blue exclusion (Figure 2f). In order to determine whether *H19* knockdown attenuated cell proliferation by inducing caspase-dependent apoptosis, caspase 3/7 assays were carried out on day 4. *H19* silencing via LNA2 triggered a significant increase in caspase 3/7 activity (*p* < 0.01) (Figure 2g). Notably, *H19* knockdown in non-neoplastic astrocytic cells did not significantly affect cell proliferation (Figure 2h). To investigate whether *H19* was involved in the migration of VUMC-DIPG-A cells, a wound healing assay was performed 24 h post-transfection. *H19* knockdown did not affect the migration of VUMC-DIPG-A cells in this experimental design (Figure 2i,j). Overall, these functional assays suggest that *H19* modulates DIPG-H3K27M cell proliferation and apoptosis.

### 2.3. Let-7a-5p Affects DIPG-H3K27M Cell Proliferation

To determine the mechanism of action behind *H19*-driven cell proliferation, we compared the transcriptome of *H19* knockdown vs. control DIPG cells. RNA sequencing analysis revealed 449 up-regulated and 622 down-regulated mRNAs upon *H19* knockdown using LNA2 in VUMC-DIPG-A cells (Figure 3a and Appendix A). Gene set enrichment analysis revealed that the most up-regulated protein-coding genes were associated with tumour necrosis factor alpha (TNFα) signalling, apoptosis, or KRAS signalling. Down-regulated genes were associated with epithelial and mesenchymal transition, UV response, or TGFβ signalling (Figure 3b).

Cytoplasmic lncRNAs often interact with miRNAs. Typically, these lncRNAs bind to complementary miRNAs, thereby displacing the interaction of the miRNA and its mRNAs targets; this results in increased translation of the target mRNAs [26]. We therefore decided to test the hypothesis that *H19* exerts its oncogenic function by buffering onco-suppressive miRNAs. As the first step, we searched the TARBASE dataset, and identified 11 miRNAs that are validated targets of *H19* (Figure 3c and Appendix A). Notably, the *let-7* miRNA family was highly represented in this list. Hence, we decided to focus further investigation on *let-7a-5p*, whose direct binding to *H19* has been demonstrated experimentally [27]. However, it is still unknown whether *let-7a-5p* affects DIPG cell proliferation.

To test our hypothesis, we transfected a miRNA mimic of *let-7a-5p* in DIPG cells. Transient overexpression of *let-7a-5p* led to a significant reduction in VUMC-DIPG-A cell proliferation five days post-transfection (Figure 3d). Morphologically, *let-7a-5p*-overexpressing VUMC-DIPG-A cells showed no obvious differences compared to those transfected with the negative control mimic (Figure 3e). These results suggest that *let-7a-5p* is involved in DIPG-H3K27M cell proliferation.

To identify which mRNAs are likely to be targeted by both *H19* and *let-7*, the list of *H19*-down-regulated mRNAs (622) was cross-referenced with the list of *let-7a-5p*-predicted mRNAs (1207) extracted from TargetScan 7.0 (Figure 3f). There were 64 overlapping mRNAs, but only eight of these were positively and significantly (*p* < 0.0001) correlated with *H19* expression in clinical samples using the data of pedHGGs from paediatric Cbioportal. *SULF2* and *OSMR* were the top two protein-coding genes that correlated with *H19* expression (Appendix A). *Let-7a-5p* had a predicted 6mer and 8mer binding site for *SULF2* and *OSMR*, respectively (Figure 3g). In keeping with our mechanistic model, the expression of *SULF2* (*p* < 0.01) and *OSMR* (*p* = 0.052) was reduced in *H19*-silenced VUMC-DIPG-A cells (Figure 3h). These results suggest that *H19* potentially sponges *let-7a-5p* thereby, causing the up-regulation of *let-7a-5p*-target mRNAs (such as *SULF2* and *OSMR*).

## 3. Discussion

In this study, we showed that *H19* is up-regulated in DIPGs, and that silencing *H19* decreases DIPG-H2K27M cell proliferation and increases caspase 3/7 activity. To the best of our knowledge, this is the first functional characterisation of a lncRNA in DIPGs. The oncogenic role of *H19* has been previously proposed for a wide range of malignancies, including adult high-grade gliomas [15]. In keeping with our findings, several lines of evidence showed that *H19* silencing leads to decreased cell proliferation in other malignancies; interestingly, this effect was explained by a wide range of mechanisms of action, which appear to be, at least in part, cell-specific [28,29,30].

Herein, we showed an association between *H19* expression and DIPGs bearing H3K27M. The molecular landscape of DIPGs is very heterogeneous [4]. Besides H3K27M, mutations in other chromatin regulators have been identified in DIPGs, including mutations in the *ATRX* and *DAXX* genes. These mutations are not common, but have been suggested to be associated with DIPG-H3K27M; however, their implication in DIPG progression and their link with H3K27M are poorly understood [4,21]. Other chromatin changes are controlled by dynamic post-translational modifications (PTMs) on the histone N-terminal tail, which influence the structure of chromatin and, hence, gene expression [31]. DIPG tumours bearing H3K27M induce a dramatic reduction in the global levels of H3K27me3 in DIPG cells [4]. Recently, other PTMs were identified, including H3K26me2 and H416ac [31]. Herein, we did not have the bioinformatics tools to determine the association between histone PTMs and *H19* expression, but this would be very interesting and the objective of future studies.

The origin of DIPGs is controversial, but recent research suggests that these malignancies originate from the partial differentiation of neural stem cells into oligodendrocyte precursor (OPC)-like cancer cells [4,32]. Herein, we showed that *H19* is up-regulated in OPC cells during brain development. In keeping with this finding, it has been shown that *H19* is down-regulated upon full differentiation of OPCs into oligodendrocytes [33]. Taken together, our results and previous publications suggest that *H19* may be implicated in the proliferation of OPC-like cancer cells, which are the cells of origin in DIPGs.

Mechanistically, *H19* is a cytoplasmic lncRNA that can sponge miRNAs to modulate gene expression [26]. Several miRNAs have been shown to directly interact with *H19*, including *miR-141-3p, miR-200b-3p*, and *let-7a* [34,35,36]. It is therefore likely that the effects of *H19* on normal and cancer cells are determined by the complex interplay of a pool of cell-specific miRNAs expressed in each cell type. To discover the miRNAs that mediate *H19* function in DIPGs, we crossed the list of *H19* targets with the list of miRNAs able to bind to the mRNAs down-regulated upon *H19* silencing. Based on this analysis, we decided to further investigate *let-7a-5p*, because this miRNA has been shown to interact with *H19* in other contexts [36,37,38] and to down-regulate 64 *H19*-dependent mRNAs. In keeping with our hypothesis, overexpression of *let-7a-5p* led to a reduction in DIPG cell proliferation; this observation suggests that *H19* buffers the onco-suppressive *let-7a-5p*, thereby increasing DIPG cell proliferation. However, *let-7a-5p* overexpression was not as cytotoxic as that of *H19* silencing; this observation suggests that *let-7a-5p* is only partially responsible for *H19* oncogenic function.

Interestingly, *SULF2* increases cell proliferation, invasion, mobility, and adhesion in MCF-7 and MDA-MB-231 breast cancer cell lines [39]. *OSMR* promotes proliferation, metastasis, and EMT in prostate cancer [40]. *OSMR* also increases mitochondrial respiration and radio-resistance in glioblastoma stem cells [41].

Cell proliferation is defined as the balance between cell survival and cell death. Our results indicate that *H19* is able to drive cell survival and suppress cell death in DIPG cells. However, *let-7a-5p* seems only to affect cell growth. Therefore, it seems likely that other miRNAs mediate the anti-apoptotic function of *H19* in DIPGs.

The primary aim of this work was to functionally characterise a DIPG-specific lncRNA, and to identify therapeutic targets for these incurable malignancies. For this reason, silencing experiments were carried out with LNAs instead of more conventional siRNAs. LNAs are more stable in biological media and are currently being investigated in clinical trials for the treatment of neurological disorders and malignancies [42,43,44,45]. Herein, we reported that an *H19*-targeting LNA (LNA2) has an in vitro IC_50_ lower than the typical maximum plasma concentrations of LNAs in clinical trials (2000–6000 ng/mL) [46]. Future in vivo studies are warranted to establish whether LNA2 is effective in clinically relevant animal models of DIPG-H3K27M.

In conclusion, this work has shown, for the first time, the role of an oncogenic lncRNA in DIPGs, which is likely to control multiple pathways in DIPG-H3K27M. We hope this work will encourage DIPG researchers to further explore the function of lncRNAs and to test their role as therapeutic agents.

## 4. Materials and Methods

### 4.1. Bioinformatic Analysis

#### 4.1.1. R2 Platform Analysis

H19 lncRNA was queried in the R2: Genomics Analysis and Visualization Platform (https://hgserver1.amc.nl/cgi-bin/r2/main.cgi, accessed on 31 December 2020) to assess its clinical relevance in DIPG tissue and normal brain tissue. The datasets used for this analysis were GSE26576 (R2 ID: ps_avgpres_gse26576geo37_u133p2, referred to as “Paugh”) and GSE50021 (R2 ID: ps_avgpres_gse50021geo45_ilmnht12v4, referred to as “Allis”) [23,24]. Data were sorted based on “disease_type” (Paugh) and “cell_type” (Allis). Only DIPG and normal brain tissues were compared.

#### 4.1.2. Paediatric Cbioportal Analysis

The paediatric Cbioportal analysis was carried out with *H19* lncRNA (https://pedcbioportal.kidsfirstdrc.org/, accessed on 31 December 2020) to assess the difference in the expression of *H19* in paediatric high-grade gliomas. Data were obtained from the publicly available Institute of Cancer Research (ICR) dataset from 2017 [25] and sorted based on either (1) the type of histone modification (wild type (WT); H3.1K27M and H3.3K27M, grouped as H3K27M), (2) brain location (hemispheric, midline, or brainstem), or (3) histological classification (DIPGs or other paediatric high-grade gliomas, including anaplastic gliomas and glioblastomas). Data were also used to perform Kaplan–Meïer analysis sorted based on the median expression of *H19* in Graphpad Prism 7 (Graphpad Software, San Diego, CA, USA).

#### 4.1.3. Single-Cell Analysis

Open access dataset on single-cell analysis of the midbrain in development (http://linnarssonlab.org/ventralmidbrain/, accessed on 31 March 2021). *H19* was queried and its expression in human cells was analysed.

### 4.2. Cell Culture

Primary human foetal cortical normal astrocytes were a gift from Prof. DK Male (The Open University, Milton, Keynes, UK) and were cultured as described elsewhere [47]. The VUMC-DIPG-A (H3.3 K27M) cell line was kindly provided by Dr. Esther Hulleman (VUMC Cancer Center, Amsterdam, the Netherlands) [22]. SU-DIPG-IV (H3.1 K27M) cells were kindly provided by Dr. Michelle Monje (Stanford University, California, USA). [48]. VUMC-DIPG-A cells were cultured in 1:1 DMEM-F12 and Neurobasal-A cell media and supplemented with a 1% (*v*/*v*) glutamax supplement, a 1% (*v*/*v*) antibiotic–antimycotic solution, 10 mM HEPES, a 1% (*v*/*v*) MEM non-essential amino acid solution, and 1 mM sodium pyruvate (Thermofisher, Paisley, UK), hereafter referred to as TBM medium. VUMC-DIPG-A medium was also supplemented with 10% heat-inactivated foetal bovine serum (Merk, Watford, UK). SU-DIPG-IV was cultured in TBM medium supplemented with a 2% (*v*/*v*) B27 supplement without vitamin A (Thermofisher), 20 ng/mL of bGFG, 20 ng/mL of EGF, 10 ng/mL of PDGF-AA, 10 ng/mL of PDGF-BB (Peprotech), and 5 IE/mL of heparin (Merk). Cells were grown in adherence and passaged using 0.25% (*v*/*v*) trypsin-EDTA (Merk). All cells were cultured at 37 ℃ in a humidified environment containing 5% CO_2_.

### 4.3. Analysis of Gene Expression

RNA was isolated from cells in the culture using the RNeasy Plus Mini Kit (Qiage, Manchester, UK) according to the manufacturer’s instructions. Reverse transcription was carried out using 1 μg of RNA per reaction using the High-Capacity cDNA Reverse Transcription Kit (Applied Biosystems, Life Technologies, Warrington, UK) according to the manufacturer’s instructions. For RT-qPCR analysis, 10 ng of cDNA were loaded in duplicate per sample using the Taqman Universal mastermix and primers. The Taqman primers used in this study were *H19* (Hs00399294_g1), *GADPH* (Hs02786624_g1), *SULF2*, (Hs01016480_m1), and *OSMR* (Hs00384276_m1).

### 4.4. Subcellular Localisation of H19

RNA was isolated from cells in the culture in nuclear and cytoplasmic fractions, which were separated using the PARIS Kit (Ambion, Thermofisher, Paisley, UK) according to the manufacturer’s instruction. For validation and localisation studies, RT-qPCR was used with the probes *MALAT1* (Hs00273907_s1), *GAPDH* (Hs02786624_g1), and *HPRT1* (Hs02800695_m1).

### 4.5. LNA Reverse Transfection

Knockdown studies were performed using the reverse transfection method. Cells were seeded in a 6-well (180,000 cells/well) or 96-well plate (5000 cells/well) with a lipid/LNA mixture prepared using Oligofectamine (Invitrogen, Thermofisher, Paisley, UK) as per the manufacturer’s protocol. The final LNA concentrations were 100 nM. All LNAs were designed using the Qiagen LNA design tool and the targeting sequences were ACTAAATGAATTGCGG (LNA 1), AATTCAGAAGGGACGG (LNA 2), and GACTTAGTGCAAATTA (LNA 3). Control LNA were also purchased from Qiagen.

### 4.6. Cell Proliferation Assay

LNA-transfected cells were seeded on 6-well plates and grown at specific times, as specified in the figure legends. On the day of analysis, cells were trypsinised, centrifuged, and resuspended in HBSS for cell counting using the trypan blue (Merk) exclusion method with either a cell haemocytometer or an automatic cell counter (LUNA, Logos Bioscience, Villeneuve-d’Ascq, France).

### 4.7. Caspase 3/7 Assays

Cells were plated in a white, flat-bottomed 96-well plates and treated with 100 nM of LNAs, as previously described. On day 4 post-transfection, the Caspase-Glo reagent was added to each well (Promega, Southampton, UK) according to the manufacturer’s instruction and incubated for 1.5 h. Luminescence was then quantified using the BMG polarSTAR plate reader (BMG Labtech, Aylesbury, UK).

### 4.8. Wound Healing Assay (Migration)

VUMC-DIPG-A cells were reverse transfected for 18 h with 50 nM LNA2, as these did not yield a significant reduction in viable cells as determined by cell proliferation assays. After 18 h, a scratch was made in the cell monolayer using a sterile P20 pipette tip. Cells were imaged periodically across the wound until the wound was closed. Images were analysed using the ImageJ (ImageJ Software, Madison, WI, USA).

### 4.9. RNA Sequencing Analysis

VUMC-DIPG-A cells were reverse transfected with 100 nM of LNA and RNA isolated 48 h post-transfection. NGS libraries were prepared according to the Ion AmpliSeq Library Kit Plus’ user guide (Thermo Fisher Scientific, Paisley, UK). For each sample, 100 ng of total RNA was reverse transcribed using the SuperScript VILO cDNA Synthesis Kit (Thermo Fisher Scientific). The resulting cDNA was then amplified for 10 cycles by adding PCR Master Mix and the AmpliSeq Human Transcriptome Gene Expression Kit panel, targeting over 20,000 genes (>95% of the RefSeq gene database). Amplicons were digested with the proprietary FuPa enzyme to generate compatible ends for barcoded adapters ligation. The resulting libraries were purified using AmpureXP beads (Agencourt, Beckman Coulter Life Sciences, Nyon, Switzerland) at a bead-to-sample ratio of 1.5× and eluted in 50 µL of low TE buffer. The libraries were then diluted 1:10,000 and quantified by qPCR using the Ion Universal Quantitation Kit (Thermo Fisher Scientific). Individual libraries were diluted to a 50 pM concentration, combined in batches of eight libraries, loaded on an Ion 540™ chip using the Ion ChefTM instrument and sequenced on an Ion S5™ instrument (Thermo Fisher Scientific). QC was manually performed for each sample based on the following metrics: Number of reads per sample >8 × 10^6^, with valid reads >90%. Raw data were processed automatically on the Torrent ServerTM and aligned to the reference hg19 AmpliSeq Transcriptome fasta reference.

The AmpliSeqRNA plugin was used to determine valid matches to the amplicon target regions in the panel using the hg19 AmpliSeq Transcriptome bed file. Each sample had more than 5 × 10^6^ total aligned reads. At least 10 mapped reads per amplicon in more than three samples were used to filter out lowly covered amplicons. Two analysis pipelines were used separately. Log-fold changes reported by DESeq2 (10.1186/s13059-014-0550-8) were shrunk with apeglm (10.1093/bioinformatics/bty895), available within the DESeq2 Bioconductor package (package version: 1.30.1, 10.18129/B9.bioc.DESeq2). Genes were reported as significant with log-fold changes >0.58 and s-values <0.01. Analyses were performed in R (version: 4.0.3, R Core Team (2020) using RStudio interface (version 1.3.1073)).

Biological function gene ontology was analysed with Gene Set Enrichment Analysis (http://www.gsea-msigdb.org/gsea/msigdb/index.jsp, accessed on 30 April 2021) and the “Hallmark” feature using genes grouped as up-regulated (log2FCs >0.58 and *s*-values <0.01) or down-regulated (log2FCs <−0.58 and *s*-values <0.01).

### 4.10. miRNA Mimic Studies

VUMC-DIPG-A cells were reverse transfected with 30 nM negative control (NC; scramble control) or the *let-7a-5p* mimic using the method previously described. The sequences used were obtained from Thermo Fisher and were as follows: *let-7a-5p* mimic (MC31809) and NC mimic (4464058).

### 4.11. Analysis of miRNA Expression

For miRNA analysis, RNA was isolated using miRNeasy (Qiagen) according to the manufacturer’s instructions. Reverse transcription was carried out using the miRNA cDNA Synthesis Kit (Applied biosystems, Paisley, UK) according to the manufacturer’s instructions using the reverse transcription primers provided with *let-7a-5p* (000377) or *U6* snRNA (001973) using 10 ng of RNA per reaction. Then, 1.7 ng of the resulting cDNA was used for the qPCR analysis using the qPCR taqman probes provided in the kits described for *let-7a-5p* or *U6* according to the manufacturer’s instructions

### 4.12. In Silico Prediction of miRNA Target Genes

Identification of the experimentally validated miRNA–lncRNA binding site was carried out with the online open access lncBase v.3 database (http://carolina.imis.athenainnovation.gr/diana_tools/web/index.php?r=site%2Ftools, accessed on 30 September 2020) [49]. Only miRNAs whose binding has been previously demonstrated by luciferase assay were taken into account. Prediction of miRNA–mRNA interactions was investigated using the open access TargetScan database (http://www.targetscan.org/vert_72/, accessed on 30 September 2020). The predicted target mRNAs were overlapped with mRNAs down-regulated upon *H19* knockdown (RNA-seq) in VUMC-DIPG-A cells. Top mRNAs that were positively correlated (Spearman values <0.05) with *H19* expression in the pedHGG clinical dataset (paediatric Cbioportal, ICR 2017 pedHGG study) were prioritised for validation.

### 4.13. Statistical Analysis

All data are presented as means  ±  SDs (standard deviations). The number of independent experiments (*n*) with replicates are specified in each legend. Normality was assessed with Shapiro–Wilk test (*p* = 0.05). Means were compared using unpaired or paired two-tailed *t*-tests for single comparison and one-way or two-way ANOVAs for multiple comparisons. ANOVA analysis was followed by post-hoc analysis. Specific post-hoc analysis is specified in each figure legend. All tests were performed using the statistical software GraphPad Prism 7 (GraphPad Software, San Diego, CA, USA). A *p*-value <0.05 was considered to be statistically significant.

## Figures and Tables

**Figure 1 ijms-22-09165-f001:**
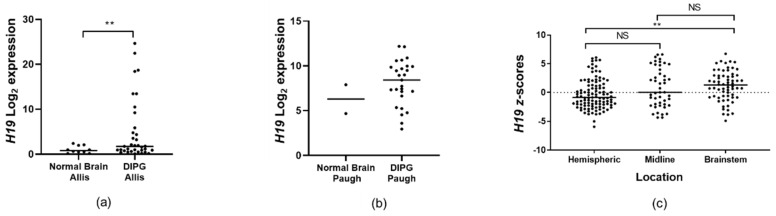
*H19* is associated with DIPGs (**a**) Clinical expression data of *H19* in the “Allis-45-custom-ilmnht12v4” dataset analysed on the R2 platform for normal brain (*n* = 10) and DIPG tissues (*n* = 35) [23,24]. (**b**) Clinical expression levels of *H19* in the “*Paugh-37-MAS5.0-u133p2*” dataset analysed on the R2 platform for normal brain (*n* = 2) and DIPG tissues (*n* = 27). (**c**) *H19* expression in patient samples with pedHGGs located in either hemisphere (*n* = 107), midline (*n* = 46), or brainstem (*n* = 65) analysed with the paediatric Cbioportal platform [25]. (**d**) *H19* expression in patient samples with pedHGGs either non-mutated/WT (*n* = 118) or carrying a H3K27M mutation (*n* = 83) analysed with the paediatric Cbioportal platform. (**e**) Expression of *H19* in histologically confirmed DIPG samples (*n* = 65) or other pedHGGs (*n* = 153). (**f**) Kaplan–Meïer curve for overall survival time (months) of DIPG patients with either high or low levels of *H19* (median expression cut-off). (**g**) Violin plot showing the Bayesian posterior probability of *H19* expression in different cell types during neurodevelopment. Data in (**a**,**d**,**e**) were analysed by an unpaired two-tailed *t*-test with Welch’s correction. Data in (**c**) were analysed with one-way ANOVA and Dunnett’s post-hoc test for multiple comparison. All data are shown as mean ± SD. Significant values are ** *p* < 0.01, *** *p* < 0.001. NS = non-significant.

**Figure 2 ijms-22-09165-f002:**
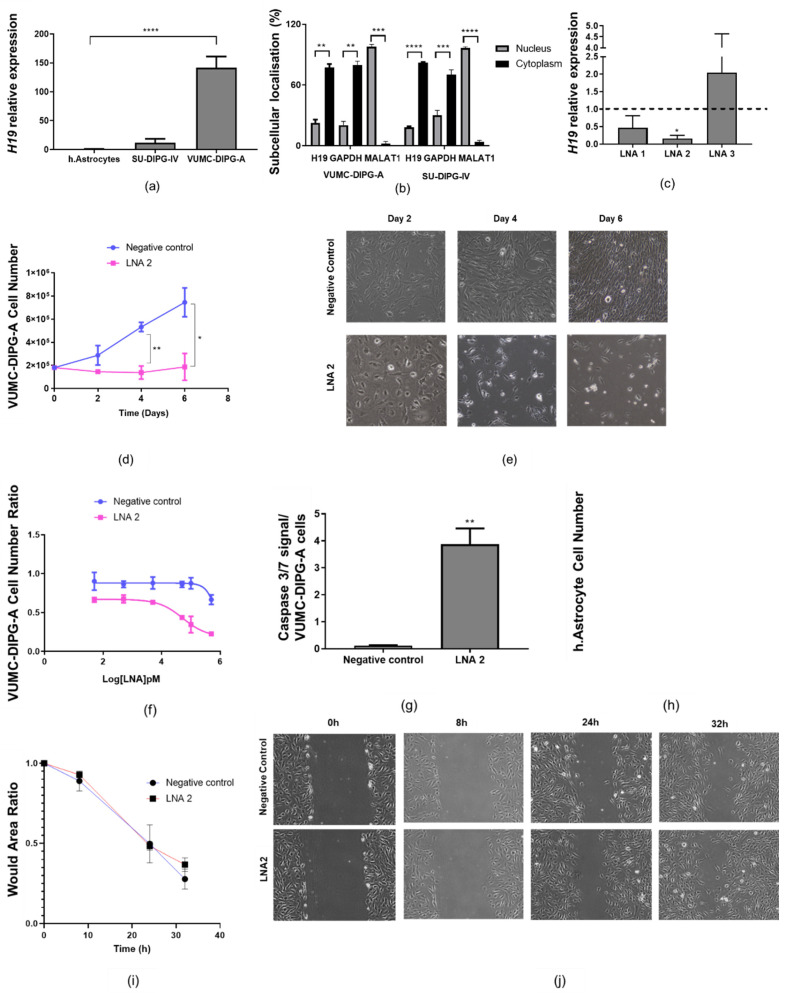
*H19* is required for DIPG cell proliferation but not migration. (**a**) Expression (qPCR) of *H19* lncRNA in a panel of DIPG cancer cells and a non-neoplastic human astrocyte cell line (*n* = 3). (**b**) Subcellular localisation of *H19* in VUMC-DIPG-A and SU-DIPG-A cells. Internal control included GAPDH (cytoplasm control) and MALAT1 (nuclear control) (*n* = 2 and *n* = 3, respectively). (**c**) *H19* RNA relative expression on LNA-silenced VUMC-DIPG-A cells (LNA1, LNA2, and LNA3) to negative control LNA-silenced VUMC-DIPG-A cells (*n* = 3). (**d**) Trypan blue exclusion cell number of VUMC-DIPG-A after 2, 4, and 6 days post-transfection with LNA2 or negative control LNA (*n* = 3). (**e**) Representative brightfield image of LNA2- or negative control LNA-transfected VUMC-DIPG-A cells after 2, 4, and 6 days. Magnification 10×. (**f**) Dose response (0.05 to 500 nM) effect of LNA7 silencing on VUMC-DIPG-A cell number (*n* = 2). (**g**) Cell viability-adjusted activity of caspase 3/7 after 4 days of LNA2-mediated *H19* knockdown in VUMC-DIPG-A cells (*n* = 3). (**h**) Cell number of normal human astrocytes after transfection with either LNA2 (50 nM) or negative control LNA on day 4 (*n* = 3). (**i**) Wound healing assay showing wound area ratio closing at 8, 24, and 32 h post-scratch (*n* = 3). (**j**) Brightfield representative images of wound healing assay. Magnification 10×. Data on (**a**) was analysed with a one-way ANOVA followed by Dunnett’s multiple comparison test. Data on (**b**–**d**) were analysed with a two-way ANOVA with Sidak’s post-hoc test for multiple comparison. Data are shown as mean ± SD. Significant values were * *p* < 0.05, ** *p* < 0.01, *** *p* < 0.001, or **** *p* < 0.0001.

**Figure 3 ijms-22-09165-f003:**
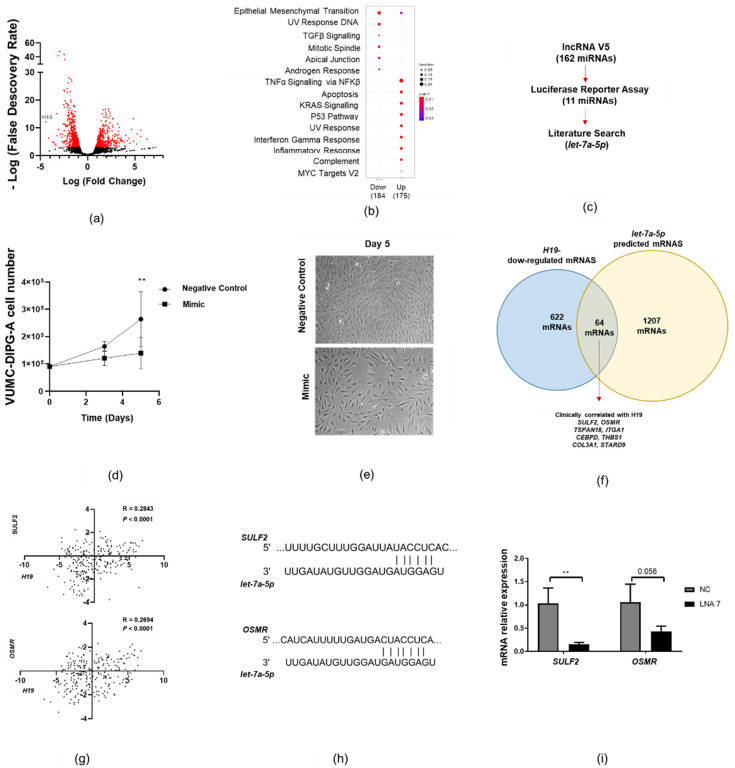
*let-7a-5p* reduces DIPG-H3K27M cell proliferation and migration. (**a**) Volcano plot of RNA-seq expression on *H19*-silenced VUMC-DIPG cells (red dots are *p* < 0.01). (**b**) Gene set enrichment analysis of up- and down-regulated (log2FC < or >0.58) mRNAs upon *H19* silencing in VUMC-DIPG-A cells. (**c**) Process diagram for the identification of previously validated *H19* miRNA targets with the lncRNA v5 database and a luciferase binding assay. (**d**) Trypan blue exclusion cell number on *let-7a-5p*-overexpressing VUMC-DIPG-A cells on days 3 and 5 post-transfection. (**e**) Brightfield images of the *let-7a-5p* mimic and the control (scramble oligonucleotide) overexpressing VUMC-DIPG-A cells (*n* = 4). Magnification 10×. (**f**) Venn diagram of cross-referencing *H19*-silenced mRNAs with *let-7a-5p*-predicted mRNA targets. Overlapping mRNAs were analysed for clinical correlation with *H19* in paediatric Cbioportal. (**g**) Correlation analysis of *H19* expression and *SULF2* or *OSMR* in the ICR 2017 pedHGG dataset on the paediatric Cbioportal platform [25]. (**h**) Graphical representation of predicted binding site between *let-7a-5p* and *SULF2* (top) or *OSMR* (bottom) mRNA. (**i**) *SULF2* and *OSMR* expression in *H19*-silenced VUMC-DIPG-A cells 2 days after transfection, analysed by qPCR (*n* = 3). Data on (**b**) were analysed with a two-way ANOVA and Sidak’s post-hoc multiple comparison. Data on (**i**) were analysed with an unpaired two-tailed *t*-test. Data on (**g**) were analysed using Spearman’s correlation. Data are shown as mean ± SD and ** *p* < 0.01.

## Data Availability

Additional data are available upon reasonable request via e-mail.

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
