# Peer review of "The Long Non-Coding RNA H19 Drives the Proliferation of Diffuse Intrinsic Pontine Glioma with H3K27 Mutation"

_ijms, 2021, doi:10.3390/ijms22179165_

Round 1

Reviewer 1 Report

Diffuse intrinsic pontine glioma (DIPG) is an incurable paediatric malignancy. Authors investigated the oncogenic role of the development-associated H19 lncRNA in DIPG, and found that H19 expression was higher in DIPG vs normal brain tissue and other pedHGGs. 

This paper is interesting. However, following points should be taken into consideration.

It is well known that H3.3K27M mutant tumors are associated with shorter survival. As mono-methylation of H3K27, H3K79, and H4K20 are all linked to gene activation, whereas tri-methylation of H3K27 and H3K79 are linked to repression. H4K20me, which is functional in DNA repair, represents a binding site for the 53BP1 protein; H3K9me3 and H4K20me3 represent epigenetic markers important to the function of 53BP1 in non-homologous end joining (NHEJ) repair. It is not clear whether other epigenetic signatures of DIPG, including H3K36me2 and H4K16ac, showed similar H19 expression of H3.3K27M mutant tumors.    

Author Response

We thank the reviewer for the suggestion and information provided regarding the DIPG molecular landscape. We have amended the discussion to suggest the reader that other associations are possible and future work is needed to fully elucidate the interactions between this lncRNA and different histone post-translational modifications. (row 237 – "Here we showed an association of H19 expression with DIPG bearing H3K27M. The molecular landscape of DIPG is very heterogeneous (4). Besides H3K27M, mutations in other chromatin regulators have been identified in DIPG. This included mutation in ATRX and DAXX genes. These mutations are not common but have been suggested to be associated with DIPG-H3K27M, however, their implication in DIPG progression and H3K27M link are poorly understood (4,21). Other chromatin changes are controlled by dynamic post-translational modifications (PTMs) on the histone N-terminal tail, which influence the structure of chromatin, hence, gene expression (31). DIPG tumors bearing H3K27M induce a dramatic reduction of global levels of H3K27me3 in DIPG cells (4). Recently, other PTMs have been identified including H3K26me2 and H416ac (31). Here, we did not have the bioinformatics tools to determine the association between histone PTMs and H19 expression, but this would be very interest and the objective of future studies.”)

Reviewer 2 Report

Although the authors address the novelty of the present study and cancer model including lack of data on diffuse intrinsic pontine glioma, the manuscript has many discrepancies and I am really concern on the  representation and quality of the study.

The concerns:

1) Figure 1 lacks statistical data on normal tissues (1a,b), therefore, p values are inadequate; In addition, in fig1d, the sample size of H3G34R is to low to conclude significantly different levels between groups.

2) Survival analysis show no difference between low and high H19 groups, however the authors argue that H19 is required for DIPG cell viability. Please, discuss it more. 

3) in figure 2e - it is impossible to inspect the difference; what is the casp3/7 signal, when other LNAs are used, does it correlate with h19 expression? Brightfield images for 2i are missing.

4) in figure 3, why do the authors need pathway analysis? Does it correlate with mirna analysis? 

5) overall, please improve figure quality.

Author Response

Please find also attached the manuscript to check the overall figure quality.

1)

We appreciate the point suggested by the reviewer, which is likely caused by the fact of how difficult is to obtain normal brain tissue for this type of comparison studies. We agree that Figure 1.b lacks of enough data points to draw any statistical significant conclusion. Therefore, figure 1.b and text has been amended to remove statistical analysis. Regarding Figure 1.a, normal tissue dataset has 10 data points, for this reason, we consider a comparison with DIPG tissue within the same dataset can be carried out.

Regarding figure H3G34R, this dataset has been removed and conclusion have been drawn from comparing paediatric gliomas with wild type (wt) and H3K27M mutated histone only.

2) 

We understand the reviewer concern regarding the link of H19 with DIPG cell viability. DIPG is a very aggressive cancer with an invariably short survival time. This might be one of the reasons why no difference in overall survival was observed. Other explanation is likely to be caused by the highly heterogenic molecular profile observed in DIPG tumours and the need for better subgrouping of DIPG tissue. This argument was already proposed by a recent study evaluating the therapeutic outcome of different available DIPG therapies based on overall survival rate. Authors discussed that “the lack of an observed difference should not be interpret as evidence of a lack of effect of these interventions: the estimates are indirect, have wide confidence intervals and moderate unexplained heterogeneity. (PMID: 28681244)” Here, we observed that targeting H19 with anti-sense oligonucleotides decreased DIPG cell proliferation, increased casp3/7signal and affected cellular pathways related to cell viability (e.g. apoptosis and cytokine signalling pathways). Future research will evaluate the translational effect of targeting H19 using mouse models of DIPG and will help to elucidate further the link of H19 with DIPG overall survival time.

3) 

We apologized for the small images provided, pictures have been cropped and their size have been increased to ease data interpretation. Similarly, brightfield images for 2i have been added (see 2j)

We appreciate the reviewer question regarding the effect of other LNAs on casp3/7 signal. However, the other LNAs screened showed no significant changes in H19 levels in comparison to negative control transfected DIPG cells.

4)

One of the aims of this article is to inspire other researchers to investigate lncRNA (and H19 specifically) functions in DIPG, hence the pathway analysis data give further guidance. Indeed, lncRNAs have been shown to affect different pathways (PMID: 27070700).

The pathways analysis was carried out using the genes deregulated upon H19 silencing. The analysis did not include miRNA expression. However, some of the identified targets of H19 and let-7 are known modulators of some of the pathways identified. For example, SULF2 has been shown to affect Epithelial-to-Mesenchymal Transition and apoptosis in hepatocellular carcinoma (PMID: 33889573)

5) 

All figures have been reviewed. Font, size and style of legends corrected and overall quality have been improved to the best of our knowledge.

Round 2

Reviewer 2 Report

The authors have improved the study according to my concerns, therefore I suggest to proceed the manuscript further.